# Revisiting the fall of the Veramin meteorite

Dan Holtstam[1], Ataollah Hassani[2]

[1]Department of Geosciences, Swedish Museum of Natural History, Box 50007, SE-104 05 Stockholm, Sweden
[2]History Department, Shahid Beheshti University, 1983969411 Tehran, Iran

*Correspondence to*: Dan Holtstam (dan.holtstam@nrm.se)

**Abstract.** The Veramin meteorite, believed to have fallen in 1880, near Varamin, Tehran province, Iran (then Persia), is one of few witnessed falls of a mesosiderite, a rare type of stony-iron meteorite. In this review, it is described that historical records show inconsistencies regarding the fall, and consequently, the naming of the meteorite. The earliest printed account, by Ferdinand Dietzsch in 1881, reported that the meteorite fell near the village "Karand" east of Tehran, with a thunder-like sound. The Shah had ordered an examination of it. Later, meteoricist Aristides Brezina named it "Veramin". Further historical accounts include descriptions by Iranian official Mohammad Hassan Khan Sani' od-Dowlah and the explorer Sven Hedin. A key document is a Persian text on a cardboard, preserved with the main meteorite mass in Tehran's Golestan Palace. Members of the nomadic Shahsevan-e Baghdadi tribal confederacy, who had winter settlements west of Tehran, are reported as eyewitnesses. The geologist Henry A. Ward provided a detailed description in 1901, confirming the meteorite's composition and securing a larger mass for analysis and distribution to museums. The exact location and date of the fall remain uncertain due to imprecise and conflicting sources. The most likely impact field is the Booghin-Eshtehard area west of Tehran, with the event happening sometime in the period February to April 1880. The original mentioning of "Karand" is a confusion with Zarand(ieh), 70 km to the west of Varamin.

## 1 Introduction

A mesosiderite is a rare kind of stony-iron meteorite, about half of which consist of iron-nickel metal and the remaining part is composed of silicates plus minor accessory minerals. They normally have a highly brecciated character and are one of the most enigmatic groups of differentiated meteorites in terms of their origin (Rubin, 1996; Benedix et al., 2014). Only seven witnessed falls of a mesosiderite have been reported to date. The Veramin meteorite that according to tradition (Ward, 1901) fell in Persia in 1880 in the vicinity of Veramin (Varamin, Tehran Province; 35°19' N, 51°39' E) belongs to this category. The purpose of the present contribution is to summarize the state of knowledge concerning the circumstances of the fall, for the international community. By using the historical method and scrutinizing the scientific literature and other written sources, inconsistencies with respect to details of the event were encountered by us and by earlier authors (available mostly in Persian publications ).

## 2 Previous work

Hassanzadeh (1986) described in some detail the petrology, mineralogy and fall history of the Veramin meteorite, in the Persian language. He made attempts to find the main mass of the meteorite that had been lost for nearly a century and finally succeeded to investigate it. One of the key contentions in his study is the questionable reliability of historical accounts regarding the timing of the meteorite fall, coupled with evident carelessness in recording its precise location. Given the incomplete and contradictory information available, he concluded that further investigations were required (Hassanzadeh, 1986).

The history of Veramin was more recently summarized in an article for the general audience by Torabi (2010) that is essentially based on the work of Hassanzadeh (1986), but with a few observations of its own.

**3 Information on the meteorite and the fall from the early sources**

The earliest printed account of this meteorite was given by the German mining engineer Ferdinand Dietzsch, who was invited to an audience with the Shah of Persia, Naser al-Din (of the Qajar Dynasty; Fig. 1), in the first half of May in 1880 (Dietzsch, 1881). According to the recounted information from the Shah, the meteorite had fallen "*in der Nähe des Dorfes Karand, 12 Meilen östlich von Teheran*" (German: "in the vicinity of the village Karand, 12 German miles [90 km] east of Tehran") with a thunder-like sound. Hassanzadeh (1986) also quotes this passage, but erroneously gives the distance as "…20 km east of

Tehran…".

The weight was given as 45 kg by Dietzsch (1881; converted from old Turkic/Persian 15 *Tabriz batman*) – a slight underestimation considering later information – of which he got 400 g with him for examination. He specifically noted a blackish fusion crust on parts of the fragments, confirming the meteoritic origin.

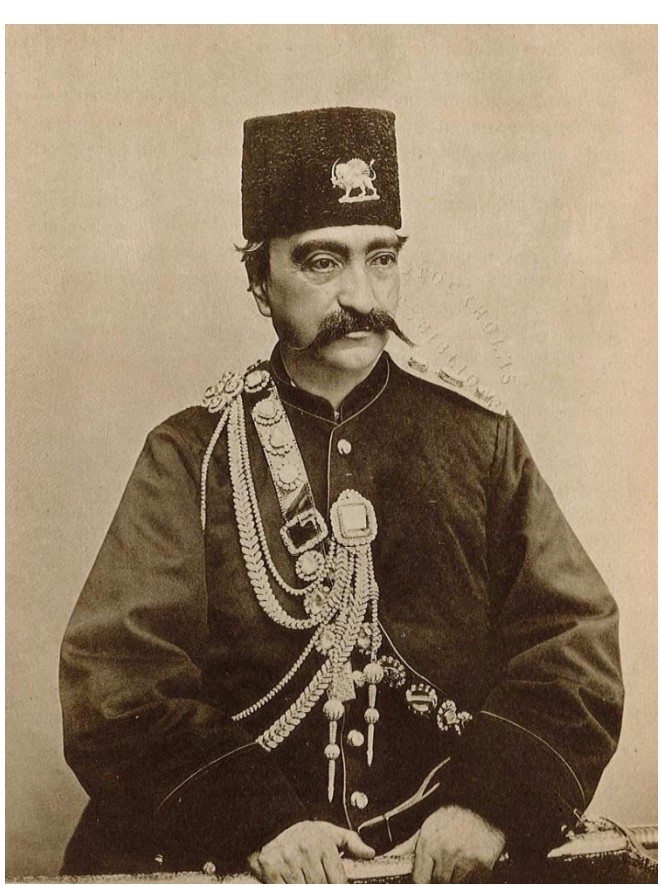

**Figure 1: Naser al-Din Qajar, Shah, was the ruler of Persia at the time of the meteorite fall and received the stone in his palace and initiated investigations of it. Photo reproduced from Hedin (1891).**

According to documents available, it was the Shah himself who asked Aligholi Khan Mokhber od-Dowlah, Vazir-e Mokhaberat (the Minister of telecommunications) to arrange a meeting with a foreign expert to examine the stone, and the minister replied in a letter that he will bring *Monsieur Dij* [Dietzsch] to the court the following day (Fig. 2).

The first investigation of the meteorite by a meteoriticist was by Aristides Brezina, who named it "Veramin" but commented that the circumstances of the fall were still shrouded in obscurity (Brezina, 1881). According to his information, the main mass

of the meteorite was kept outdoor, in the royal gardens in Tehran. He also noted the occurrence of a fusion crust, but mentioned the fact that some rust had developed on his specimen. From the primary mineralogy, Brezina (1881) grouped the meteorite with the then newly fallen (in 1879) Estherville and other known mesosiderites.

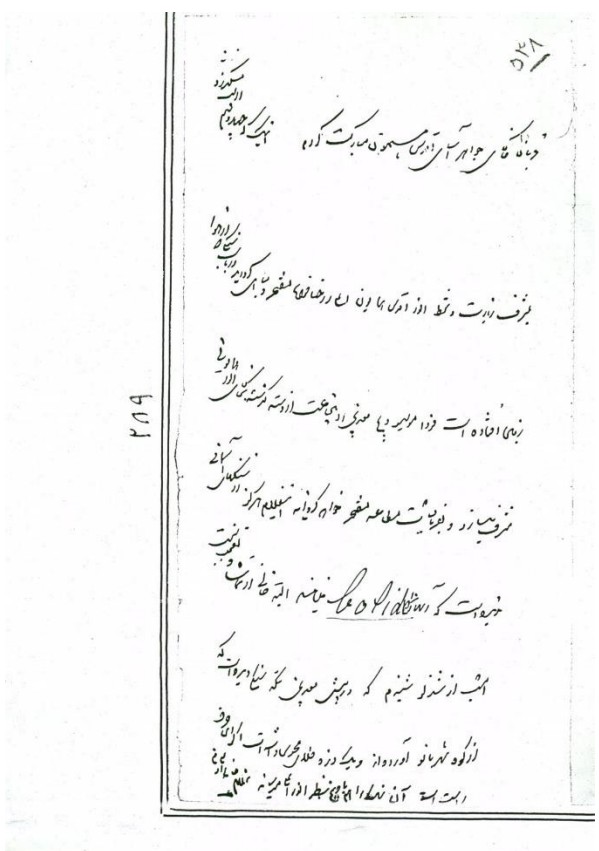

**Figure 2. The Letter of Minister Aligholi Khan Mokhber od-Dowlah to Naser al-Din Shah, saying that that he will bring Monsieur Dietzsch, a German mining engineer, to the royal court the following day (May, 1880). Reproduced from original archival document.**

The first written description of the meteorite was, however, in an at the time unpublished note by Mohammad Hassan Khan

Sani' od-Dowlah (1843–1896) – from 1304/1887 E'temad os-Saltanah – a Qajar scholar and statesman, who was contemporary with the event, and the first Iranian historian to report it (E'temad os-Saltanah, 1367/1948). Nearly two months after the meteorite fell, he wrote in his memoirs, referring to investigations carried out: "…its weight is approximately 15 *man* of Tabriz. This time, they tested the said stone and it turned out that it contained four tenths of iron in pure form, and in addition to iron, a metal known as nickel was found in it, and for this reason, the stone looked bright with silvery particles. Its pure iron grains

are mixed with a black mineral called aluminum silicate.. Green crystals [pyroxene] have also been observed in it, similar to which can be seen in most celestial stones." (For a general description of the mineralogy of Veramin, see, e.g., Hassanzadeh et al., 1990). Sani' od-Dowlah was probably influenced here by an oral report of Dietzsch, although he did not mention him. The explorer Sven Hedin visited Persia in the spring of 1890 as an interpreter in a Swedish-Norwegian diplomatic mission. At one occasion, on May 28[th], he was admitted to the museum of the royal palace (Hedin, 1891). Only the Shah himself and his

Minister of Treasury, who was the key holder, had access to it. On the floor, close by the entrance, was the meteorite lying on a pedestal. It was said that just one year before it was moved from the palace yard where it had been kept beside a small pond. A piece of paper with Persian writing was available with the meteorite in the museum (Hedin's translation into Swedish): *I närheten af Bugin och Echtehard hörde man i Hedschras 1298:de år plötsligen ett mycket starkt dån, som kom ifrån molnen och förskräckte dem, hvilka hörde det. Man hörde på samma gång nio starka knallar, liknande kanonskott och kommande från*

*molnen. Man såg därpå någonting komma ifrån molnen, som liknade rök och eld och, då det föll till jorden, borrade det ned sig två meter. De personer, som voro tillstädes, togo upp aeroliten från jorden. Den väger 45* [sic] *batman (= 45 kg.)* [Translation from Swedish: "Near Boogin and Eshtehard, in the 1298[th] year of Hijri, a very loud roar was suddenly heard coming from the clouds, which frightened those who heard it. At the same time, nine loud bangs, similar to cannon shots and coming from the clouds, were heard. Then something that looked like smoke and fire was seen coming from the clouds, and

when it fell to the ground, it buried itself two meters deep. The people who were present picked up the meteorite from the ground. It weighs 45 [sic] batman (= 45 kg)."

Hedin had the intention to hunt for additional fragments of the meteorite in the field. However, he could only get vague and contradictory information from locals, on both the time and the place of the fall and thus the plans for a search expedition were shelved. He had brought with him a small piece of the meteorite, borrowed from Adolf Erik Nordenskiöld, the famous Swedish

explorer, to make comparisons if something was to be found (Hedin, 1891).

The most detailed description dealing with this meteorite was provided by the American geologist and meteorite collector Henry A. Ward (1834–1906), who travelled to Persia in an effort to see it and if possible, to obtain a specimen (Ward, 1899; 1901). Ward´s interest came to be after having met Nordenskiöld in Stockholm in 1898, who had shown him a piece of the meteorite. He claimed that Nordenskiöld had received it "from a returned Swedish servant (barber) of the Shah". However,

according to specimen labels (Fig. 3) preserved at the Swedish Museum of Natural History, the source of the meteorite samples was Dr. Bertrand Hybenette (1846–1930), a Swedish dentist who for many years was employed as physician-in-ordinary to the Shah (Svenskt Biografiskt Lexikon, 1971). Hybenette had obviously passed on fragments of it to Nordenskiöld and the museum in Stockholm in 1883. According to Lindström (1884), the museum had in total 407 g of Veramin in its possession, listed as gift from the Shah of Persia. The mass available today is 347 g, arguably the second largest sample amount outside

Iran (cf. Grady, 2000).

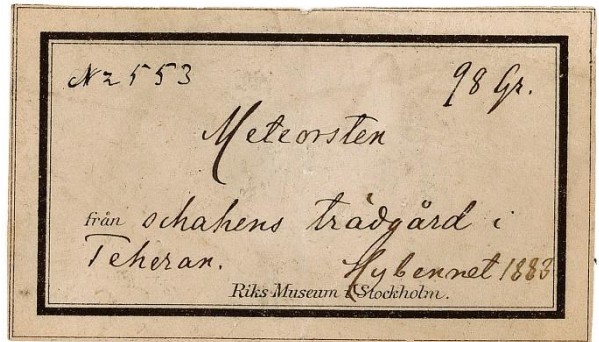

**Figure 3. Original specimen label, Swedish Museum of Natural History, representing a part of one of the larger sample masses outside Iran. "Meteor stone from the garden of the Shah in Tehran. Hybennet 1883". Reproduced from original archival document.**

In Tehran, Ward had an encounter with the Shah (successor of the previous mentioned) and got the permission to remove a larger piece, ca 1.3 kg, that was used for analysis and his description of the meteorite (Ward, 1901), and from which fragments subsequently have been distributed to several museums in the world. According to a weighing carried out at the occasion, the total mass was 51.5 kg. In his paper, he reproduced a Persian text, which was found on a cardboard sign attached to the

meteorite in the museum. The original text in Persian, mentioned by both Hedin and Ward, was in the past noted as lost (Hassanzadeh, 1986, 1990). For this reason, Hassanzadeh (1986) used the translation of Ward (1901) and transcribed it back into Persian. One limitation of this approach is that retranslating an already translated text may compromise the authenticity and credibility associated with the original. Torabi (2010) was, however, able to get access to the original text in 2007, at the Golestan Palace (Kakh-e Golestan) Museum, Tehran, but did not provide a new transcription of it. The cardboard text is still

(June 2023) on display with the meteorite (Fig. 4). The translation of Ward (1901; carried out by a Mr. Edward Tyler, American consul in Iran), is given here [with omitted parts compared to the present cardboard, and our clarifying interpretations, in square brackets]:

[1.      He is Allah The Almighty and The Invincible]

[2.]     On the 8th of Jamadi ul-av[v]al A.H.1298, [*luu yil*], at a place called Boog[h]in, the winter quarters of the Bag[hd]adi

Shahsevan, about three hours before sunset,

[3.] there appeared in a clear sky, between Boog[h]in and Esht[e]hard, a small cloud, followed immediately by an appalling noise.

[4.] The people were greatly moved, and rushed to their tent doors. While standing there in expectation,

[5.] they heard nine more terrific explosions, like to the reports of a cannon.

[6.] Following upon these, something resembling smoke [and fire] proceeded from the cloud, clearly visible to all the people,

[7.] descended to the earth and buried itself [two cubit, ~ 2 m] in the ground. A shepherd near the place

[8. who had narrated the event], with fear and trembling pointed it out to the people. Some of them went

[9.] and turned up the ground and took out the stone. [His highness] Hadayat ullah khan, Kajar [Hedayatollah Khan

Qajar], the son of the late Eesa [Isa] khan [the former] Beglerbegee [Beyglarbeygi of Tehran],

[10.] Governor of the tribe, took possession of the stone, and reported

[11.] the occurrence to [the exalted trustees of government], and sent the stone [to Dar al-khelafat al-Bahera = "The bright city where the seat of the Caliph is"].

There is also a note added to the main text of the cardboard label: "On the 21st of Rajab ul-morajjab 1316 A.H. (December 6, 1898 A.D.), *It yil* (Turkic dog year), Professor Ward came from America and weighed the stone which was seventeen and half [*man* of] Tabriz [equal to 52.5 kg; however, according to Ward (1901): "… its weight was just 113 1/2 pounds (~ 51 2/5 kg)], and photographed it."

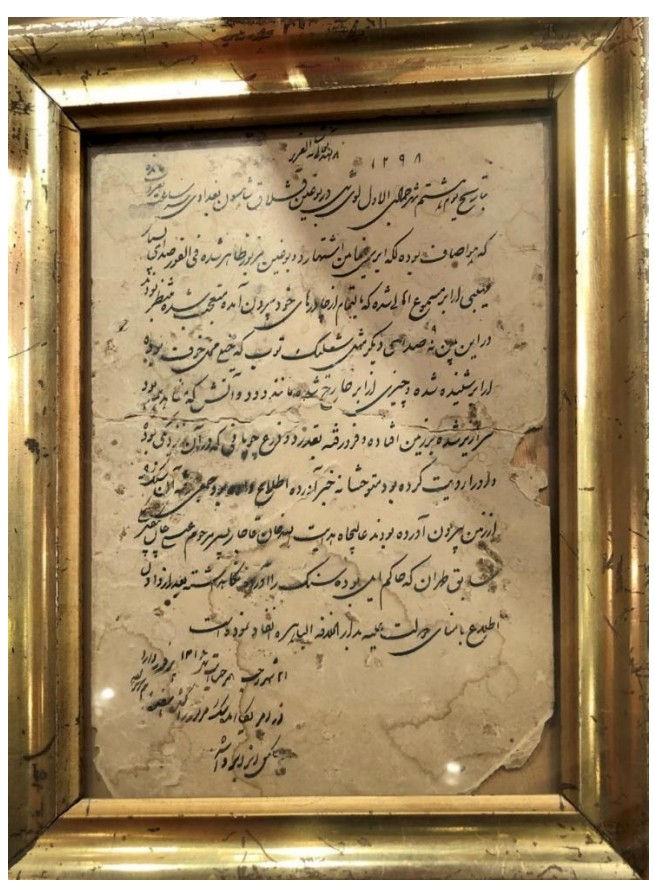

**Figure 4. The cardboard text to the main mass of the Veramin meteorite, in Persian. Palace museum, Golestan Palace, Tehran. Photo: A. Hassani.**

## 4 The place of the meteorite fall

Some mistakes occur in the translations (and Hedin's version is obviously abridged), but the original text itself is interesting. The indicated place of the event, Booghin (also Buqin; 35°35'N, 50°38'E, 1300 m above sea level, in present Malard County, Tehran Province), a small area in the historical Zarand district with some tiny villages like Asad abad-e Booghin, Mehdi abad-e Booghin, etc., is at a linear distance of 97 km from Varamin (Fig. 5). Notably, Torabi (2010) also identified the probable fall location as Buqin, based on the translation of Hassanzadeh (1986) of the cardboard text.

Ward (1901) was reluctant to change the name of the meteorite partly because he was not able to locate the place names from the cardboard text on a map, and added: "…as the palace people called it Veramin, the same name by which Diet[z]sch noted the stone [incorrect statement] a few months after it fell, it seems proper that his name should be retained. Veramin is a small plain in the District of Karand some fifteen miles eastward of Teheran…." (Ward, 1901). Dietzsch (1881), however, never mentioned "Veramin", but referred to "Karand", which seems to be a mishearing of Zarand. The Zarand district (nowadays Zarandieh County) was part of the old Saveh province. Veramin (Varamin), on the other hand, was one of the five districts that made up the city of Tehran at the time of the meteorite fall, and had itself four sub-districts (Houtum-Schindler, 1897). None of these borders the area of the fall indicated above. There is an additional, independent European source as regards the fall, Dr. Joseph Désiré Tholozan, head court physician of the Shah (reported by Daubrée, 1884): "*La chute a eu lieu dans le district de Zerind, à 100 km à l'ouest de Téhéran*" [French: "The fall took place in the district of Zerind, 100 km west of Tehran."] Despite the odd spelling of Zarand, this is probably the most correct notion (the distance on roads is close to 100 km, from Golestan Palace to Booghin).

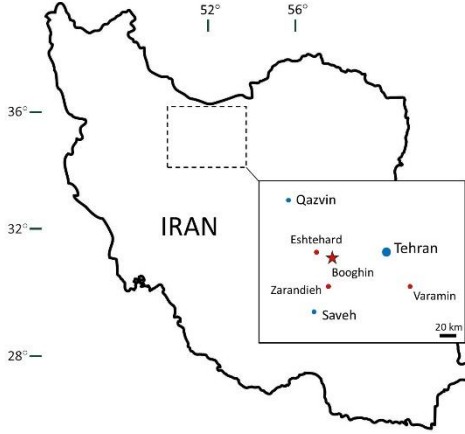

**Figure 5. Map of Iran, with the key localities mentioned. The meteorite likely fell near Booghin, located nearly 100 km from Varamin (Veramin). Drawing by authors.**

Shahsevan-e Baghdadi is a confederacy of tribes consisting of 2 big branches: Lak with 14 tribes, and Arıkhlu wıth 15 tribes, that were residing in a vast area of Central Iran (Hassani, 1400/1980). It is a fact that they had winter settlements along the roads west of Tehran, including the Booghin-Eshtehard area. The Arıkhlu-Kalavand and some tribes of Shahsevan-e Inanlu have been permanently settled and are still living there today, e.g., Qara Torpaq [Topraq], Qozlu, Qanlı quyu, Alvird hisari, Gandeh quyu, Qılıç kandi, Qishlaq-e Hosein abad, Qishlaq-e Musalu, Mehdi abad-e Booghin, etc. (Mohit, 1392/1972: 69-70). The exact location of the meteorite impact remains unknown, but it is clear that it should be sought in the Booghin-Eshtehard area, according to the information in the Iranian sources. The erroneous information about the location, Veramin, is traced to Brezina, who possibly received it from Baron Emil Gödel-Lannoy, an Austrian diplomat who supplied the meteorite samples for examination (Brezina, 1881).

The naming of a historical person in the cardboard text, Hedayatullah Khan Davalu-ye Qajar, who is said to have been involved in the delivery of the stone, is interesting. (Davalu/Devalu or Divalu was one of the six tribes of the Qajars; meaning "those who have camels".) On the cardboard label, he is named as "the governor of the tribe", which is incorrect. When the meteorite fell in the in Zarand of Saveh, Kamran Mirza, Nayeb os-Saltanah, a son of Naser al-Din Shah, was in charge of ruling Saveh, Zarand, and Shahsevan-e Baghdadi, and the capital city of Tehran. Muhammad Mahdi Khan E'tezad od-Dowlah had assumed the position of "the ruler of Qom and Shahsevan-e Baghdadi" on behalf of Kamran Mirza, and after a while, the rule of Saveh was added to his position. At the same time, Kamran Mirza had appointed Hedayatollah Khan as "Nayeb al-hukumah-ye Shahsevan-e Baghdadi" [the deputy ruler of Shahsevan-e Baghdadi]. This is known from a letter written by Kamran Mirza to Hedayatollah Khan, two months after the meteorite fell (National Library and Archives of I.R. Iran. Document No. 296/24162). Hedayatollah Khan Davalu-ye Qajar's paternal grandfather and father were both "Beyglarbeygi-ye Dar ol-Khelafeh-ye Tehran" [the high-rank ruler of the capital city of Tehran].

**5 The time of the fall**

According to the cardboard text, the 8th of Jamadi-ul-avval A.H. 1298 is given as the date of the fall, which corresponds to April 7, A.D. 1881. This is an impossibility, because Dietzsch saw the meteorite in Tehran already in May 1880. Sani' od-Dowlah did not mention any specific day or month, but gives the year of the meteorite fall as 1297 A.H. (E'temad os-Saltanah, 1367/1948: 2008). Hassanzadeh (1986, 1990) relied on Sani' od-Dowlah's report and suggested that the date should be corrected to 8th Jamadi ul-avval 1297 A.H. (April 18, 1880). Interestingly, both Sani' od-Dowlah and the anonymous writer of the cardboard text states the Turkic zodiac year for the event as *luu yil* (dragon year), which would correspond to the period March 20, 1880 – March 20, 1881. Then followed the *yïlan yil* (snake). Obviously, the cardboard text is in error as regards the lunar year, written as 1298 A.H.

In various sources, the months of February (Daubrée, 1884), April (Brezina, 1881; Lindström, 1884; Hassanzadeh, 1990) and May (Ward, 1901) have been inferred as the time of the meteorite fall. We know that Dietzsch and his companions visited the Shah in the first half of May 1880 (Dietzsch, 1881). It is reasonable to assume that the whole process of excavation and removal of the stone from the ground, placing it in the hands of Hedayatollah Khan, informing the sovereign's court of the event, transporting the meteorite a distance of about one hundred kilometres to Tehran on animal, and the initial investigations carried out by of Naser al-Din Shah and courtiers, has altogether been time-consuming. In addition, the pastoral tribesmen mentioned would have left the region for *yaylaq* (summer highland pasture) several weeks before. Therefore, it is concluded the meteorite could not have fallen in May, A.D. 1880.

The calendar of yearly moving of the nomads of the region can be helpful in further clarifying the date. From the last days of March to the beginning of April, the nomads lead their cattle from the winter quarter pens, *qishlaq*, to open space, and after a short stay in temporary pastures, they enter the ways to the distant summer pastures in the Qaraqan Mountains. Accordingly, February is a month of dwelling in the winter quarters and April is the month of moving to the summer quarters. The information from Tholozan (Daubrée, 1884) indicated a date around 15th of February. This period is acceptable, both from local geography and considering the moving calendar of the Shahsevan-e Baghdadi tribes. However, it is in conflict with the notion of *luu yil*, which started on March 20. The later part of March A.D. 1880 is thus another possibility.

According to the cardboard text, the event happened about 3 hours before sunset, which for the period February–April in the region would be between 4 and 5.30 pm.

## 6 Conclusions

The sound and light phenomena reported in the sources quoted are typical for a meteoroid (fireball) entering the atmosphere. Light effects were probably less prominent in this case because the event happened in broad day-light. The air bursts are produced when the original body is broken up into fragments while experiencing extreme pressure and temperature conditions during descent through the atmosphere (Lissauer and De Pater, 2013).

From the description, it not unlikely that several pieces landed on the ground, and they may well be still partly preserved considering the climatic conditions (arid) of the Booghin area and that they, like the only known specimen, might have been quite deeply buried and protected. So far, search efforts that may have been undertaken have been misled by the notion of "Veramin", pointing to the wrong area.

There is no first-hand information as regards the fall available. The purported eyewitnesses of the Shahsevan-e Baghdadi people are mentioned in the cardboard text. There is a number of secondary and, in details, contradictory sources that we have taken part of. It appears as the simplest explanation to the mess is largely the ignorance of the Westerners at that time, about the language and geography of Persia. The vagueness in the local sources is another contributing factor. An alternative, but unlikely, explanation could be that someone deliberately spread false information about the fall to conceal it. This, however, seems less plausible since the stone was already secured before the foreigners became involved, and it was the Shah himself who decided to inform them. The meteorite's nearly century-long obscurity that followed likely resulted from it being stored among less important objects in the palace museum after Henry Ward's visit—an outcome seemingly tied to the political instability during the waning years of the Qajar dynasty, the subsequent rise of the Pahlavi regime, and an overall lack of interest at the time.

*Author contribution*: D.H. initiated and conceptualized the study. A.H. has investigated documents in Iranian archives. The authors wrote the paper in close cooperation.

*Competing interests*: The authors declare that they have no conflict of interest.

*Acknowledgements*: We thank H. Pourkhorsandi and J. Hassanzadeh for careful and constructive reviews.

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
