# Peer review of "Revisiting the fall of the Veramin meteorite"

_History of Geo- and Space Sciences, 2024_

## Author Response (AR2)

Dear Editor

We are most thankful for the careful review work carried out on our manuscript.

Detailed responses to comments by reviewer J. Hassanzadeh are given here:

Line 31: The phrase "(available in Persian publications only)" is not accurate, as it overlooks the date-correcting note published by Hassanzadeh in the journal Meteoritics in 1990.

This has now been rephrased.

Line 41: I suggest replacing "lost for many years" with "lost for nearly a century," since Ward's 1901 paper was the last documented mention of the stone's whereabouts before its rediscovery in 1984, as narrated by Hassanzadeh (1986). Meteorite reference books (e.g., Mason, 1962) that mention the Veramin meteorite typically list only Tehran as its location. Notably, based on personal communication with staff at the General Office of Palaces, Ministry of Economic Affairs and Finance of Iran, the stone's century-long obscurity appears to have stemmed from it being placed in storage among unidentified or unrelated objects after Ward's visit— likely a result of political instability during the decline of the Qajar dynasty, the subsequent transition to the Pahlavi regime, and a general lack of interest. Including this historical context would significantly enrich the narrative of a paper that takes a historiographic approach.

We have changed to "lost for nearly a century" according to the recommendation.

We have tried to expand with a brief comment on the reasons for the long-forgotten meteorite, following the suggestion here, under "Conclusions".

Lines 205–206: Hassanzadeh (1990) is the appropriate English-language reference for the corrected fall date of April 18, 1880, and should also be cited here.

This reference is in place.

Additionally, Hassanzadeh later conducted a detailed geochemical analysis of the Veramin mesosiderite at the University of California, Los Angeles, with results published in Geochimica et Cosmochimica Acta (Hassanzadeh et al., 1990). These data could inform the brief discussion of composition in the present manuscript. On this note, I would also recommend revisiting the mineralogical description: the large green crystals referred to in line 80 are pyroxenes, not olivine, and feldspar is only present as microscopic grains as purely calcic plagioclase.

It is changed to "pyroxene". It remains unclear what is meant by "aluminium silicate" in the original text. We have added the paper as general reference to the composition of Veramin.

Given that the two main questions addressed in the current paper—namely the fall date and location—were already investigated in Hassanzadeh (1986), the title might more accurately reflect the nature of this contribution by including the word "Revisiting" (e.g., "Revisiting the Fall Date and Impact Site of the Veramin Meteorite").

In the first review round of the paper, we were advised to include "review" in the title to emphasize the character of the paper. But the present suggestion works as well, in our opinion. We use a simplified version here: "Revisiting the fall of the Veramin meteorite".

/Dan Holtstam & Ataollah Hassani